# A Brief Review of Machine Learning-Based Bioactive Compound Research

**Jihye Park [1,†], Bo Ram Beck [2,†], Hoo Hyun Kim [1], Sangbum Lee [3,*] and Keunsoo Kang [1,*]**

[1] Department of Microbiology, College of Science & Technology, Dankook University, Cheonan 31116, Korea; apjh1998@dankook.ac.kr (J.P.); hoohyun.kim@dankook.ac.kr (H.H.K.)

[2] Deargen Inc., 193, Munji-ro, Yuseong-gu, Daejeon 34051, Korea; brbr777@deargen.me

[3] Department of Software, College of Software Convergence, Dankook University, Yongin 16890, Korea

[*] Correspondence: sblee@dankook.ac.kr (S.L.); kangk1204@dankook.ac.kr (K.K.); Tel.: +82-31-8005-3224 (S.L.); +82-41-550-3456 (K.K.)

[†] These authors contributed equally to this work.

**Abstract:** Bioactive compounds are often used as initial substances for many therapeutic agents. In recent years, both theoretical and practical innovations in hardware-assisted and fast-evolving machine learning (ML) have made it possible to identify desired bioactive compounds in chemical spaces, such as those in natural products (NPs). This review introduces how machine learning approaches can be used for the identification and evaluation of bioactive compounds. It also provides an overview of recent research trends in machine learning-based prediction and the evaluation of bioactive compounds by listing real-world examples along with various input data. In addition, several ML-based approaches to identify specific bioactive compounds for cardiovascular and metabolic diseases are described. Overall, these approaches are important for the discovery of novel bioactive compounds and provide new insights into the machine learning basis for various traditional applications of bioactive compound-related research.

**Keywords:** bioactive compound; natural product; machine learning; bioinformatics; cheminformatics; chemical space; cardiovascular disease; metabolic disease

## 1. Introduction

Bioactive compounds are chemicals that exist in trace amounts in natural products (NPs) from plants and animals. They are used as components of health supplements or as early substances in drug development [1–3]. Bioactive compounds are considered to have suitable scaffolds that can interact with various proteins in living organisms through evolutionary pressure. Therefore, they were frequently used as initial substances for a significant number of therapeutics [1,4–6]. For example, plants contain rapidly evolving, multifunctional metabolic enzymes that appear to influence the production of structurally diverse chemicals such as secondary metabolites [7]. In addition, many approved drugs are structurally similar to bioactive compounds [8], providing a clue that finding bioactive compound analogs may be the right strategy for novel drug development. For example, a clinically approved anti-mitotic drug, eribulin (Halaven), is derived from the natural product halichondrin B [9]. In addition, a survey estimated that 28% of all currently approved drugs are either NPs or their derivatives [10]. In fact, a major source of oral drugs is derived from NPs [11]. Therefore, the discovery of novel bioactive compounds in NPs or their derivatives has routinely been a way to develop valuable therapeutic agents.

Chemical space is defined as a virtual space, comprising all theoretically possible small organic molecules in excess of $10^{60}$ [12–14]. Previously, navigating the entire chemical space was thought to be impractical due to mostly technical limitations. However, with the theoretical and practical innovation of machine learning (ML) fields such as deep

learning (DL), which is undergoing rapid development in support of hardware such as graphics processing units (GPUs) and tensor processing units (TPUs) in recent years, it has become possible to navigate a chemical subspace with desired properties. There are many successful case studies of the application of ML for various purposes. For example, recent studies identified many NP-like small molecules through the exploration of the chemical space of NPs [15–17]. In addition, ML models can accurately discriminate wild from farmed salmon based on gas chromatography with flame ionization detector (GC-FID) fatty acid profiles [18]. Therefore, with the help of state-of-the-art machine learning techniques, a variety of tasks such as discovering novel bioactive compounds with desired properties by training data measured through various large-scale experiments have become possible.

Recently, many ML algorithms are being applied to drug discovery, including hit identification, lead optimization, and clinical development stages [19–21]. Their goal is to accelerate the discovery of drugs more quickly and efficiently. For example, the traditional hit identification process requires labor-intensive, massive high throughput screening (HTS) that consumes many resources. To overcome this process, ML- or DL-based drug-target interaction (DTI) algorithms are used to identify small molecules that can bind to the desired target proteins using either sequence-based or 3D structure-based data [22–25]. To evaluate the properties of a given compound, cheminformatics approaches, such as the quantitative structure–activity relationship (QSAR), a computational modeling method for predicting the relationship between structural properties and biological activities of chemical compounds, are used widely, along with the prediction of absorption, distribution, metabolism, excretion, and toxicity (ADMET) [26–28]. Collectively, these computational predictions are effective at reducing the time and cost required for drug development, and many appropriate applications can also be applied to NP-based drug or active molecule discovery, which is expected to accelerate such processes.

In this review, we avoid discussing in-depth characteristics of machine learning techniques and the discovery and/or development of novel bioactive compounds, but rather offer an overview of recent research trends toward the machine learning-driven prediction and evaluation of bioactive compounds based on actual cases.

## 2. Cheminformatics, Bioinformatics, and Databases for Machine Learning

Cheminformatics is a research field that achieves desired goals by utilizing various information on chemicals, including 2D and 3D structures of chemical species. In general, the structural information of chemicals can be standardized and converted to machine-readable formats such as a simplified molecular-input line-entry system (SMILES) [29]. SMILES strings can be stored electronically into a database and then utilized through a variety of computational algorithms to identify, evaluate, and predict properties of chemicals. For example, the PubChem database [30] currently contains more than 100 million unique chemical structures extracted from contributed PubChem substance records (https://pubchemdocs.ncbi.nlm.nih.gov/statistics (accessed on 10 December 2021)). There are other databases for chemical collections such as ChEMBL (https://www.ebi.ac.uk/chembl/ (accessed on 10 December 2021)) [31] and ZINC15 (https://zinc15.docking.org/ (accessed on 10 December 2021)) [32]. Therefore, integrating cheminformatics methods and databases into existing drug discovery procedures has become indispensable for successful drug development. Accordingly, it is not at all surprising that many initial hits (a hit is defined as a molecule that shows the desired type of activity in an experimental screening assay) and further optimization for lead compounds can be performed via computational methods, such as computer-aided drug design (CADD) approaches. There are many successful applications of CADD in the development of currently available commercial drugs. For example, zanamivir (Relenza), a neuraminidase inhibitor that is used as a prophylaxis to treat the symptoms of influenza, was approved by the United States Food and Drug Administration (FDA) in 1999 [33] and is one of the most successful outcomes of CADD.

The identification of new bioactive compounds for a given target protein requires not only chemical information, but also detailed molecular information about the target protein, such as amino acid sequence, domain, and three-dimensional structural information. Fortunately, this kind of information has been accumulated experimentally and computationally over several decades [34]. Conceptually, genome (DNA sequences), RNA sequences, protein sequences, and chemical structure information that is fundamentally similar between individuals within the same species can be defined as static information. On the other hand, molecular information that can be dynamically changed between individuals and/or conditions, such as DNA sequence variants (DNA mutations), gene expressions, and protein amounts, can be defined as dynamic information (Figure 1). Rapid innovation in high-throughput screening methods and large-scale expansion into virtually all molecular and clinical domains has dramatically expanded the scale of this molecular data over the past decade [35]. Therefore, bioinformatics has emerged to analyze this vast amount of molecular data.

The terms "cheminformatics" and "bioinformatics" are used interchangeably as the two fields deal primarily with computer-readable data from small chemical compounds, metabolites, RNAs, DNAs, and/or proteins. As mentioned above, cheminformatics mainly focuses on various chemical information, while bioinformatics primarily deals with DNA, RNA, and protein information. Many DNA and RNA nucleotides and/or protein databases are publicly available, including the nucleotide database maintained by the National Center for Biotechnology Information (NCBI) (https://www.ncbi.nlm.nih.gov/nucleotide/ (accessed on 10 December 2021)) [36], the UniProt database (https://www.uniprot.org/ (accessed on 10 December 2021)) [37], and the Protein Data Bank (PDB) database (https://www.rcsb.org/ (accessed on 10 December 2021)) [38]. In addition, there are several web-based ready-to-use databases related to bioactive compounds, which have been well described in a review [39]. These databases mostly contain sequence (string) information of given molecules, obtained from various molecular experiments. For example, DNA and RNA can be represented as sequences of A, G, C, and T (or U for RNA), while protein can be denoted as strings of twenty letters where a single letter indicates an amino acid. In contrast to this static molecular information, there are big databases that contain massive amounts of dynamic molecular information generated by high-throughput screening approaches, such as next-generation sequencing (NGS). Compared to the static molecular information mentioned above, NGS-based high-throughput screening systems have been generating molecular dynamic data, such as RNA (transcription or gene expression), DNA (single nucleotide variant, SNV; insertion and deletion, INDEL; copy number variation, CNV; structural variation, SV), and DNA or RNA binding proteins in various cells, tissues, and even at the single cell level. Currently, more than 10 petabytes of raw sequencing data (https://trace.ncbi.nlm.nih.gov/Traces/sra (accessed on 10 December 2021)) can be downloaded from the Sequence Read Archive (SRA) website. The majority of these data is from gene expression profiling experiments, such as RNA-seq. For example, changes in the expression level of approximately 60,000 genes (or 230,000 transcripts) in human cells, with drug treatment, can be measured with an RNA-seq approach. With 100 samples, the total number of values would be 6,000,000 for genes (or 23,000,000 for transcripts), and this information can be represented as a matrix (Figure 1). However, since most genes are not normally expressed, the number of values will typically be reduced to less than one-third of that number. Then, the mode of action (MOA) prediction of the given drug can be achieved by analyzing this dynamic information using various bioinformatic algorithms and/or databases. In general, a typical MOA prediction can be conducted using gene ontology (GO) analysis and/or gene set enrichment analysis (GSEA). One of the most commonly used tools for interpreting transcriptome changes is the GSEA [40], which lists biological pathways that have been statistically significantly altered between different conditions. This knowledge-based information can be useful for narrowing down features (genes) for training because, in general, the number of features is much larger than the number of samples when building an ML model with gene expression

profiling data. Another obstacle to applying machine learning technology to gene expression data is that individual gene expression data have been processed in different analysis pipelines, so unknown bias (noise) that is difficult to identify, such as batch effects, different sequencing devices, and/or various experimental conditions, is inherent to the data. To minimize this bias and provide vast transcriptome resources in a user-friendly manner, a recent study constructed a transcriptome database called ARCHS4 (https://maayan-lab.cloud/archs4/ (accessed on 10 December 2021)), comprising more than 200,000 human and mouse transcripts processed through a uniform analysis pipeline [41]. In the case of DNA variants in humans, gnomAD (https://gnomad.broadinstitute.org/ (accessed on 10 December 2021)) provides an excellent and the largest source of DNA variant information identified in multiple populations via a wide variety of large-scale sequencing projects [42].

Overall, advances in cheminformatics, bioinformatics, and a variety of publicly accessible databases have emerged rapidly over the past decade, accelerating the process of interconnecting chemistry, biology, and drug development. This system will be used to expand our limited understanding of chemicals and ultimately build promising ML-based models that generate novel bioactive-like compounds, with high efficacy and low toxicity, against various diseases, including cardiovascular and metabolic diseases.

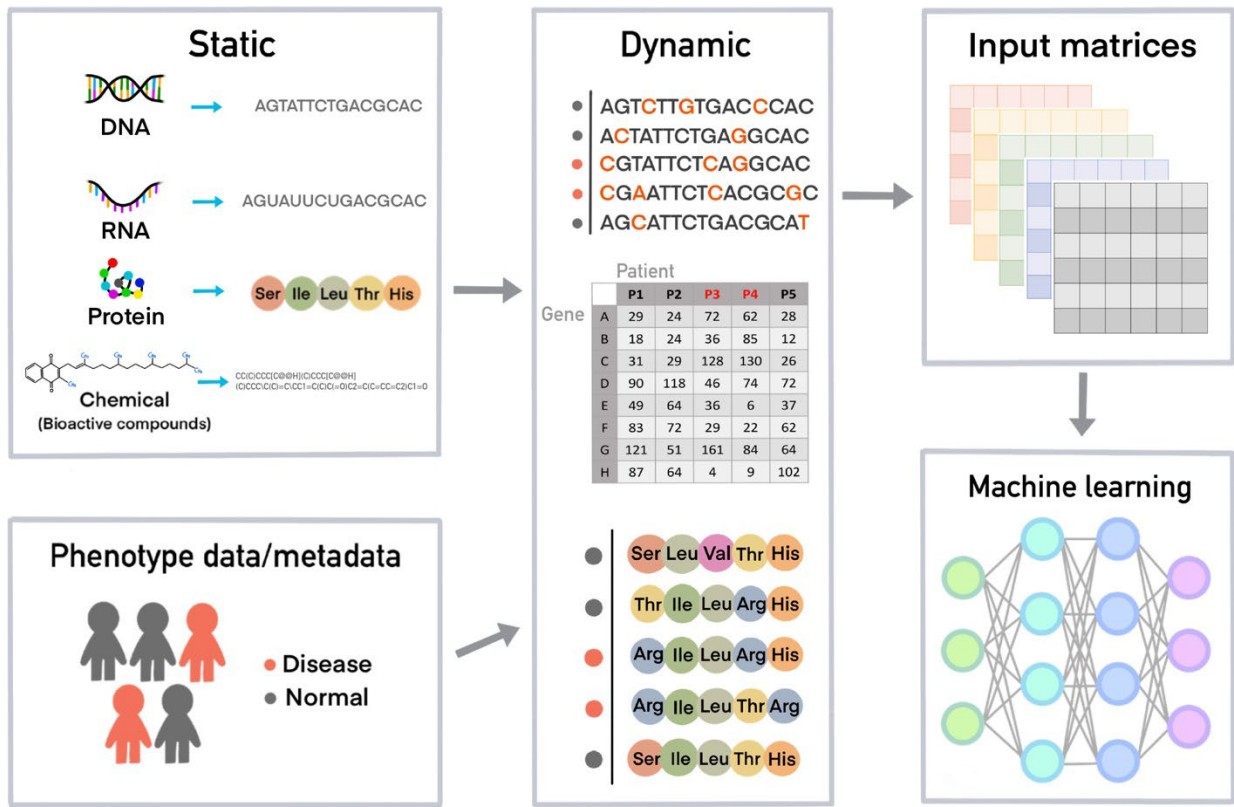

**Figure 1.** Various molecular and phenotypic data can be integrated to build a machine learning model. DNA, RNA, and protein, as well as chemical information, can be represented as strings (**top left**). This kind of fundamental molecular information can be defined as static information (deterministic), which shows little or no difference between individuals. On the other hand, phenotypic information (e.g., clinical data; **bottom left**) and high-throughput screening data (e.g., RNA-seq) produce dynamic information of molecules that can be integrated and represented as matrices (e.g., rows represent features and columns indicate samples, or vice versa) (**middle**). A collection of properly labelled matrices (**top right**) is used as input data to build a machine learning model for a particular purpose (**bottom right**).

### 3. Chemical Space Where Unidentified Bioactive Compounds Exist

Chemical space, defined as the total descriptor space that encompasses all the small carbon-based molecules that could in principle be created [43], is a multi-dimensional space in which the area to explore varies depending on what characteristics of chemicals (e.g., molecular mass, lipophilicity, toxicity, drug-likeness, number of hydrogen donors and acceptors, fraction of rotatable bonds, and druggability) the coordinates are set. In order to reach out after a subspace (e.g., NP-like chemicals) in the chemical space, it is necessary to be able to calculate unique properties of a group of similar chemicals numerically. For example, Chen et al. successfully defined a chemical space where NP-like chemicals are located and found that this space also contains chemicals highly relevant for drug discovery [44]. Fortunately, most properties of chemicals can numerically be calculated through various algorithms with metrics. For instance, the quantitative estimate of drug-likeness (QED) metric [45] estimates the drug-likeness of chemicals, and the absorption, distribution, metabolism, elimination, and toxicity (ADMET) properties of given chemicals can be calculated through various algorithms developed by computational and medicinal chemistry fields [46–53]. These numeric values, if appropriately used, will be a guide to discover novel hits for a given target protein through exploring a desired NP-like chemical space.

### 4. How a Machine Learns from Data and Creates a Model for a Task Using Machine Learning Algorithms

Machine learning (ML) is a research area encompassing computational methods of learning relationships from data without specifying them. Deep learning (DL) is a subset of ML [54] and is used in almost all areas because of the recent rapid development of hardware. In general, these approaches utilize a large number of quality-assured samples with many features. There are two types of ML methods, depending on dependent variables (or labels). Supervised learning methods learn the relationships between input features and the given labels of samples from a training dataset, whereas unsupervised learning algorithms infer patterns of input features without labels. We focus only on the supervised learning method in this review.

To build a model that can accurately classify samples into classes (classification) or predict values for samples (regression), a dataset, which contains a large number of correctly labelled-samples with many features, should be divided into three sub-datasets, called training, validation, and test datasets. The training dataset is used for training to create an optimal model among the various models to be tested. Therefore, only the training dataset is used to train the models. In contrast, the validation set is a dataset used for testing multiple models with different hyper-parameters to find the optimal values for the best model. Through extensive training steps using training and validation datasets, the best model, which has been trained and validated, is selected. Then, the selected model's classification (or regression) performance is evaluated using the test dataset. Therefore, the proper way to introduce a ML-based model is to explicitly describe all processes and parameters as much as possible [55]. Because the performance of a ML-based model is solely based on training, validation, and test datasets as well as ML algorithms with parameters (e.g., random initializations, data shuffling, dropout, etc.), it should be mandated to report all the information for the sake of transparency and reproducibility [56]. Details of machine learning have been well covered in recent reviews [57–59].

Numerous ML-based applications have been developed for a variety of purposes, including drug discovery. For example, Shin et al. introduced a transformer-based deep learning model named MT-DTI that can accurately predict binding affinities between drugs and a target protein [60] (Figure 2). Briefly, this model utilized PubChem's approximately 97 million compounds to train key knowledge such as the syntax and semantics of small molecules using their SMILES information. The trained molecular transformer (MT) information was transferred into a drug target interaction (DTI) model (so-called

transfer learning), and the final DTI model, called MT-DTI, was constructed using two DTI datasets, KIBA [61] and Davis [62]. These datasets contain experimentally validated interactions between kinases and kinase inhibitors. The MT-DTI model takes two input sequences, the SMILESs of chemicals and the amino acid sequences of a target protein, to predict their binding affinities, and it can therefore be used to identify potential target proteins of known bioactive compounds or vice versa.

Nuclear magnetic resonance (NMR) spectroscopic data can also be used to build ML-based classification models. Martínez-Treviño et al. [63] built a model that can predict NP classes of chemicals using an ML-based approach with one-dimensional $^{13}$C NMR spectroscopic data (Figure 2). In this study, the proposed approach does not rely on a typical peak matching task that requires available well-annotated NMR spectroscopic databases of chemicals, but they constructed a ML model that can predict the presence or absence of a particular NP family from the $^{13}$C NMR spectra. To this end, they trained the model with continuous variables of NP spectra in the NAPROC-13 database [64], which contains (13)C spectral information of more than 18,000 unique NP structures and found that a XGBoost-based classifier accurately predicted a given NP into eight NP families (sesquiterpenoid, diterpenoid, triterpenoid, lignan, steroid, chroman, flavonoid, and alkaloid).

String information, such as DNA nucleotides and/or protein amino acid sequences, can be used to build a ML model for a particular purpose. Walker et al. [65] developed a machine learning bioinformatics method for predicting a natural product's antibiotic activity directly from the sequence of its biosynthetic gene cluster. In this study, they assembled a training dataset from the MiBIG (Minimum Information about Biosynthetic Gene Custer) database, comprising sequences of biosynthetic gene clusters (BGCs). They assembled a dataset of known BGCs paired with the activity of their products by representing the BGCs as vectors based on the number of times various gene annotations appeared in the cluster. Then, binary classification models for antibacterial, anti-Gram-positive, anti-Gram-negative, antifungal/antitumor/cytotoxic, antitumor/cytotoxic, and antifungal activities were constructed using this information as the training data set. The proposed binary classifiers can achieve up to 80% accuracy. Another study also built a ML model with the BGC information. Hannigan et al. [66] present a deep learning strategy (Deep-BGC) that offers reduced false positive rates in BGC identification and an improved ability to extrapolate and identify novel BGC classes, compared to existing machine-learning tools. DeepBGC is a deep learning approach that uses a Bidirectional Long Short-Term Memory (BiLSTM) RNN [67] and a word-embedding skip-gram neural network (pfam2vec) such as word2vec [68]. The input consists of amino acid sequences of protein family (Pfam) domains [69] represented by vectors, and the output is a sequence of values between 0 and 1, representing the prediction score of whether a given domain is part of a BGC. This implementation improved the accuracy of BGC detection in genome sequences and the ability to identify BGCs capable of encoding natural products with novel biological activities.

The identification of novel NP-like chemicals with desired properties is of great interest in drug discovery. Grisoni et al. [70] present a machine-learning workflow for the identification of NP mimetics with the desired polypharmacology. More than 3.3 million commercially available compounds were computationally screened to identify chemicals similar to (−)-galantamine, a natural product-based drug approved by the US Food and Drug Administration (FDA) for the treatment of cognitive decline in mild to moderate Alzheimer's disease (AD) [71]. To this end, weighted holistic atom localization and entity shape (WHALES) descriptors [72], which represent the partial charge distribution and three-dimensional shape of compounds, were used to identify initial compounds similar to (−)-galantamine. Then, the compounds most similar to (−)-galantamine were further evaluated using the SOM-based prediction of drug equivalence relationships (SPiDER) [73] and target inference generator (TIGER) [74] applications, which predict target proteins of a given compound. Because (−)-galantamine reversibly inhibits acetylcholinester-

ase (AChE), the purpose of this process was to identify AChE inhibitors. Finally, this computational pipeline identified eight compounds with bioactivity on at least one of the macromolecular targets of (−)-galantamine, showing different polypharmacological properties.

Inferences about the origin of NPs provide insight into our understanding of the physicochemical properties of NPs. Pereira [75] developed a machine learning-based classification model to classify NPs into three classes: marine natural products (MNPs), terrestrial natural products (TNPs), and NPs that appear in both the terrestrial and marine environments. Briefly, molecular structures were retrieved from the Reaxys® database (Elsevier Information Systems GmbH; https://www.reaxys.com (accessed on 10 December 2021)) and were filtered appropriately. A total of 22,398 NPs were defined, which comprised 10,790 MNPs, 10,857 TNPs, and 761 as both. The data set was randomly divided into a training set of 15,676 NPs and a test set of 6722 NPs. Then, classification models were constructed using random forest, support vector machine, and multilayer perceptron networks with an overall predictive accuracy of 81% for the test set. Although all these examples only represent limited use cases of machine learning algorithms, these examples show how machine learning techniques can be utilized for NP-related research.

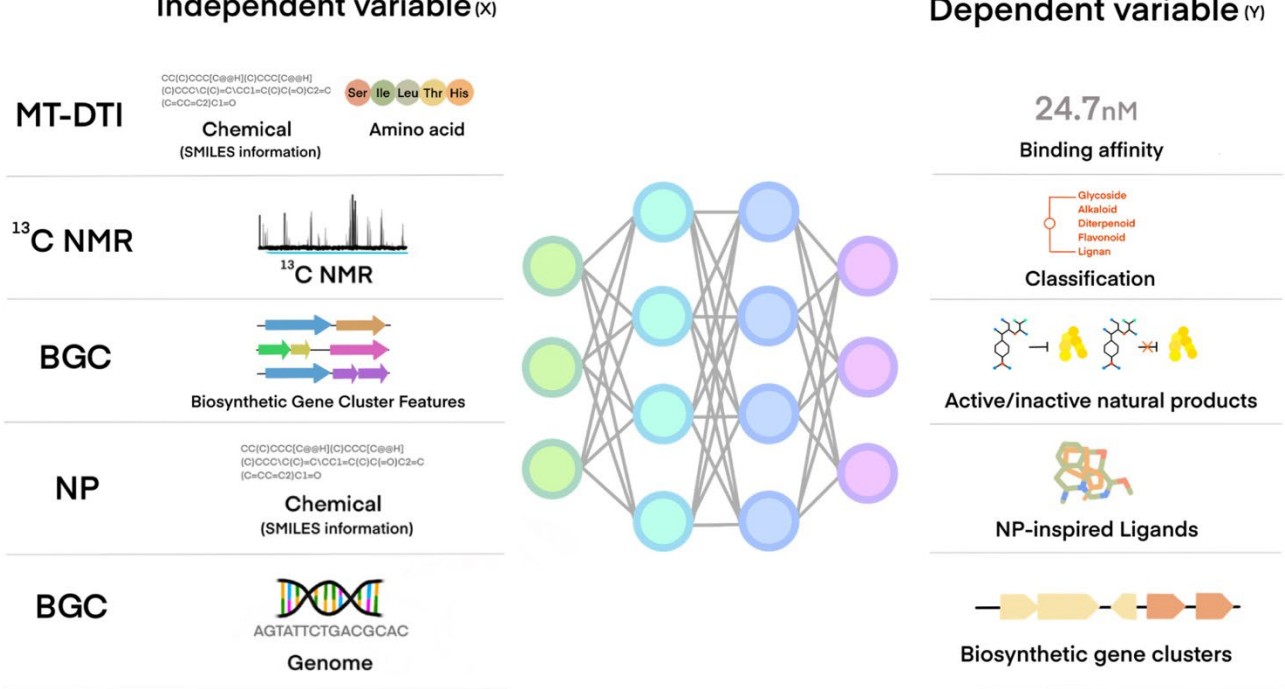

**Figure 2.** Machine learning-based models have been developed for different purposes. Various input data were used as independent variables (X, **left** panel) to predict dependent variables (Y, **right** panel). Five different examples are shown. The first example shows a model (called MT-DTI) that can predict the binding affinity (Y, the equilibrium dissociation constant known as $K_D$) between a given protein and ligand (small chemical compound). Strings of protein (amino acids) and ligand (SMILES) information (X) are used as input. The second model takes NMR data (X) as input and classifies them into known NP classes (Y). The third model analyzes genomic sequences called biosynthetic gene clusters (X) and predicts the likelihood that a natural product will have activity (Y) from those gene clusters. The fourth model takes a SMILE of a chemical of interest (X, e.g., natural product) and identifies similar small chemicals (Y) with desired multi-target profiles on disease-related targets. The last model detects biosynthetic gene clusters (Y) using a deep learning model called DeepBGC. Bacterial genome sequences are used as input (X).

### 5. Machine Learning Application of NP or NP-Like Chemical Compounds Discovery for Cardiovascular and Metabolic Diseases

Machine learning approaches may identify active molecules derived from NPs to improve human health in numerous areas of interest through methodologies described in previous sections. From a metabolic disease perspective, there are many attempts to discover NP mimetics that target specific proteins. For example, PPARγ is a master regulator protein of metabolism which is identified to be involved in metabolic diseases such as chronic inflammation, obesity, and diabetes when its function is lost or dysregulated [76,77]. Rupp et al. identified several active compound structures toward PPARγ that are related to cinnamic acid and truxillic acid through non-linear Bayesian regression with Guassian processes and virtual screening [78]. Another study suggests caffeine as an anti-diabetic NP through an iterative stochastic elimination (ISE) model built on in-house data and confirmation with wet experiments [79]. Recently, a deep learning based model using latent knowledge, molecule to target interactions, and compound property as heterogeneous input features predicted several NPs or NP-derivatives for various cardiovascular diseases (CVD): ergosterol and arginine for heart failure; reserpine, norepinephrine, octopamine, and digitoxin for hypertension; resveratrol for myocardial infarction; and aspirin and agmatine for stroke [80].

In an extensive perspective of metabolic syndrome and CVD, inflammation is a biological process that is critical in the first line of defense of the host immune system, and dysregulatory inflammatory responses can lead to cardiovascular and metabolic diseases [81,82]. A recent study [83] identified capsaicin, hypaphorine, and moupinamide among NPs as anti-inflammatory drug candidates using the ISE model, and the results were further supported by the literature. Another study employed linear discriminant analysis to identify two NP derivatives, alizarin-3-methylimino-*N*,*N*-diacetic acid and (+)-dibenzyl-L-tartrate, as anti-inflammatory agents [84]. Chagas-Paula et al. demonstrated the application of a decision tree model and a back propagation multilayer perceptron model to identify new natural products with cyclooxygenase-1 and 5-lipoxygenase dual inhibition activity from Asteraceae species extracts [85].

Although case studies for specific usage, covering predictions for wet experiments of ML methods to predict bioactive NPs, are scarce, the studies discussed above demonstrate the potential power and opportunities of ML methods in this field, supported by either literature or experimental evidence.

### 6. Conclusions

Although ML-based approaches typically achieve high performance for classification or regression tasks, there are still some limitations to be widely applied in NP-related research. First, the number of natural product-related data in which bioactivity or a target have been experimentally verified is insufficient to build an ML model. Second, while ML-based models generally outperform conventional approaches for many tasks, the reason why the model makes such a decision cannot be interpreted directly (so called "black box"). Third, an ML model is a mathematical function whose tunable parameters (values may change during training) and hyperparameters (not updated during training) must be set appropriately, but these values are iteratively tested during training to achieve the best performance, which is a time-consuming operation. Nevertheless, recent advances in machine learning have made a breakthrough in resolving these issues. For example, scientific interest in the field of explainable artificial intelligence (XAI) is emerging to explain and interpret ML models [86], while automated machine learning (AutoML) seeks to automate the entire pipeline of ML model building: automatically select, compose, and parametrize ML models [87]. Therefore, machine learning-based bioactive compound research will benefit from these state-of-the-art ML technologies. To this end, it is most important for researchers who conduct bioactive compound-related research to track newly developed ML applications and collaborate with experts in the field.

## 7. Future Perspective

As discussed in previous sections, ML technology can establish strategically faster studies and new opportunities to develop bioactive NPs. Despite the usefulness of ML methods, there are a few more topics that need to be considered for practical application.

NPs are mostly generally recognized as safe (GRAS, https://www.fda.gov/food/food-ingredients-packaging/generally-recognized-safe-gras (accessed on 10 December 2021)) and food-grade substances and used as ingredients for a variety of health supplements, functional food products, and cosmetic products. GRAS-listed NPs (i.e., berberine, canthaxanthin, eugenol, L-Glutamine, or limonene) can be directly applied to such products without any complex legal regulation, and this accessibility is a strength of NPs. In contrast, a downside of NPs is that their therapeutic efficacy is often low and limited compared to synthetic compounds. For example, phytoalexin coumarin is a plant-derived NP to defend plants from microbial infection and is studied as a bioactive NP for numerous disease indications [88]. However, coumarin itself needs to be better improved, or so-called optimized, as a derivative to achieve better effectiveness against bacterial infection, hypertension, or inflammation [88,89]. Therefore, the weight between using NPs of nature by themselves or further developing NP derivatives must be considered prior to using ML models.

To process the lead optimization task more efficiently, many machine learning approaches have been studied recently: (i) atom modification reinforcement learning models that add or delete atoms or bonds [90,91], (ii) generative reinforcement learning which generates similar but modified structure [92], (iii) generative machine learning with controlled chemical properties that also generates similar modified structure with preserved predictive properties [93], and (iv) a 3D structure-based ligand design model that uses a 3D crystal structure of protein and ligand to generate novel molecules [94]. These applications demonstrated the possibility of lead optimization or de novo small molecule-focused machine learning methods. Their goals are to apply listed models to drug discovery, but the basis of these models is not limited to de novo synthesis or the lead optimization of new chemical entities. By adopting such methods, NPs can serve as "seed" or "starting" molecules and NP derivatives as resulting modified products with preserved chemical structure similarity.

**Author Contributions:** Conceptualization, K.K. and S.L.; investigation, J.P., B.R.B., and H.H.K.; writing—original draft preparation, K.K., J.P., B.R.B., and S.L.; writing—review and editing, K.K. and S.L.; visualization, J.P. and H.H.K.; supervision, K.K.; funding acquisition, K.K. All authors have read and agreed to the published version of the manuscript.

**Funding:** This study was funded by a research grant from the Korea Food Research Institute (Project Number: E0210601-01). This work was supported by the National Research Foundation of Korea (NRF) grants funded by the Korean government (MSIT) (NRF-2019R1C1C1010385) and by a grant from the National R&D Program for Cancer Control, Ministry of Health and Welfare, Republic of Korea (1720100). This research was also supported by the Basic Science Research Program through the National Research Foundation of Korea (NRF) funded by the Ministry of Education (NRF-2020R1A6A1A03043283). The Department of Microbiology was supported through the Research-Focused Department Promotion Project as a part of the University Innovation Support Program for Dankook University in 2021.

**Institutional Review Board Statement:** Not applicable.

**Informed Consent Statement:** Not applicable.

**Data Availability Statement:** Not applicable.

**Conflicts of Interest:** B.R.B. is employed by Deargen Inc., and K.K. is a shareholder of Deargen Inc.

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
