# Peer review of "A Brief Review of Machine Learning-Based Bioactive Compound Research"

_applsci, doi:10.3390/app12062906_

Round 1
Reviewer 1 Report
Artificial intelligence and machine learning represent very actual research topics, and I consider that the topic of the manuscript is very interesting.
However, even the authors have named their article “A brief review……”, in my opinion, the quality of the manuscript would be enhanced if more information is added. The authors stated that they “avoid discussing in-depth characteristics of machine learning techniques” (lines 60-61), but I consider that a better presentation of these techniques is necessary.
Also, the article is about ML-based research of natural products as potential bioactive agents, yet I don’t consider that, throughout the article, these aspects are emphasized enough. The authors should pay more attention to this part. Also, I would consider interesting the introduction of a section about ML and QSAR analysis.
Author Response
- Artificial intelligence and machine learning represent very actual research topics, and I consider that the topic of the manuscript is very interesting. However, even the authors have named their article “A brief review……”, in my opinion, the quality of the manuscript would be enhanced if more information is added. The authors stated that they “avoid discussing in-depth characteristics of machine learning techniques” (lines 60-61), but I consider that a better presentation of these techniques is necessary.
We appreciate your critical comments. In fact, when I was asked to write a short review on this topic (”Prediction and evaluation of bioactive compounds using artificial intelligence”), I was hesitant because of the lack of ongoing ML-based research processes compared to other areas of research. Most ML-based approaches related to this topic are unfocused and scattered. So, I decided to write an overview of ML and list current research trends. As suggested, we have rewritten some parts for a better presentation.
- Also, the article is about ML-based research of natural products as potential bioactive agents, yet I don’t consider that, throughout the article, these aspects are emphasized enough. The authors should pay more attention to this part. Also, I would consider interesting the introduction of a section about ML and QSAR analysis.
Thank you for your suggestion. I agree with you. We have added an additional section about ML and QSAR in the introduction section and rewritten some parts as requested.
Reviewer 2 Report
The author shall have to modify the high resolution photos for all figures.
The author shall have to emphasize more contributions on the relevant recent works with specific discussions in the conclusion section.
The author shall have to present the conclusion section.
The author shall have to give the concrete idea for doing such kind of research logically.
Author Response
- The author shall have to modify the high resolution photos for all figures.
Thank you for the comment. The image has high resolution (300 DPI) but the resolution was decreased when it was converted to the PDF file. We will upload original high-resolution files with the manuscript this time.
- The author shall have to emphasize more contributions on the relevant recent works with specific discussions in the conclusion section.
Thank you for the suggestion. We have rewritten the manuscript as requested.
- The author shall have to present the conclusion section.
We have added the conclusion section in this revision.
- The author shall have to give the concrete idea for doing such kind of research logically.
Thank you for the suggestion. We have rewritten the manuscript as requested.
Reviewer 3 Report
Manuscript ID: applsci-1546717
Type: Review
Title: A Brief Review of Machine Learning-based Bioactive Compound Research
This review article is written in a good manner and presented the machine learning and its role in identifying the bioactive compounds. Here are my comments and questions:
Page (3), line (4): “Many nucleotide (DNA and RNA are collectively called nucleotide”
I think this phrase should be stated as “many DNA and RNA nucleotides …..”
The authors should add a section of “limitations” of ML in this review article
What is the role of algorithms in drug discovery?
Author Response
- Page (3), line (4): “Many nucleotide (DNA and RNA are collectively called nucleotide” I think this phrase should be stated as “many DNA and RNA nucleotides …..”
Thank you for the comment. This has been fixed.
- The authors should add a section of “limitations” of ML in this review article
Thank you for the comment. We have added a discussion regarding limitations of ML-based researches in the conclusion section.
- What is the role of algorithms in drug discovery?
As suggested, we have rewritten some parts for a better presentation.
Round 2
Reviewer 1 Report
Thank you for responding to my previous suggestions.
I have no other comments.
Reviewer 3 Report
The authors followed the recommendations and answered all the questions.